# *CYP1B1-RMDN2* Alzheimer's disease endophenotype locus identified for cerebral tau PET

Determining the genetic architecture of Alzheimer's disease pathologies can enhance mechanistic understanding and inform precision medicine strategies. Here, we perform a genome-wide association study of cortical tau quantified by positron emission tomography in 3046 participants from 12 independent studies. The *CYP1B1-RMDN2* locus is associated with tau deposition. The most significant signal is at rs2113389, explaining 4.3% of the variation in cortical tau, while *APOE4* rs429358 accounts for 3.6%. rs2113389 is associated with higher tau and faster cognitive decline. Additive effects, but no interactions, are observed between rs2113389 and diagnosis, APOE4, and amyloid beta positivity. *CYP1B1* expression is upregulated in AD. rs2113389 is associated with higher *CYP1B1* expression and methylation levels. Mouse model studies provide additional functional evidence for a relationship between *CYP1B1* and tau deposition but not amyloid beta. These results provide insight into the genetic basis of cerebral tau deposition and support novel pathways for therapeutic development in AD.

Alzheimer's disease (AD) is a neurodegenerative disease featuring amyloid-beta (Aβ) plaques and neurofibrillary tau tangles[1]. Aβ and tau measurements using positron emission tomography (PET) are common in research (i.e., amyloid/tau/neurodegeneration (A/T/N))[2].

Genetic factors conferring susceptibility to or protection from AD are important for identifying biological pathways for drug development and personalized medicine[3]. Large-scale genome-wide association studies (GWAS) using case-control designs have identified risk genes in immune, tau, Aβ, lipid, and other pathways[4,5]. The strongest AD genetic risk locus is *APOE* (apolipoprotein E) ε4 (*APOE4*)[6]. Large case-control studies are often limited because participant neuropathology is unknown.

Endophenotype studies complement case-control studies by testing genetic variants against disease pathology[7]. Studies have assessed genetic predictors of Aβ PET measures[8–13]. Most genetic studies of tau have utilized cerebrospinal fluid (CSF) tau measures due to non-availability of large tau PET datasets[14]. One study investigated the association of [18F]flortaucipir PET with *BIN1*, finding an association between a known *BIN1* risk single nucleotide polymorphism (SNP; rs744373) and greater tau[15]. Another performed a GWAS on tau PET endophenotypes and identified two genetic loci (*PPP2R2B* and

*IGF2BP3*), but a modest sample size (*n* = 754) and no replication sample[16,17]. Guo et al. performed a GWAS on tau PET (*n* = 543) and identified two genetic loci (*ZBTB20* and *EYA4*) associated with elevated tau accumulation and worse clinical performance[18].

Here, we perform the largest GWAS of PET-based cortical tau to date (*n* = 3046). We include data from twelve independent cohorts. We also assess the relationship of the top SNP with cognitive decline and additive and interaction effects with diagnosis, *APOE ε4* status, and Aβ positivity. We map topographic distribution of the top variant effect on voxel-wise tau deposition. We perform a gene-set enrichment analysis, assess gene expression levels in human brain tissue and single-nucleus RNA-Seq data, map the expression of the top genes in the Allen Human Brain Atlas, and perform methylation and expression quantitative trait loci (eQTL) analyses. Finally, we investigate expression levels of the top gene in tau and Aβ mouse models[19–21].

## Results

### Genome-wide association analysis (GWAS)
Meta-analyzed GWAS results from seven discovery cohorts (*n* = 1446) are shown as quantile-quantile (Fig. 1A) and Manhattan (Fig. 1B) plots. No systematic *p*-value inflation was found (genomic inflation factor

e-mail: asaykin@iu.edu

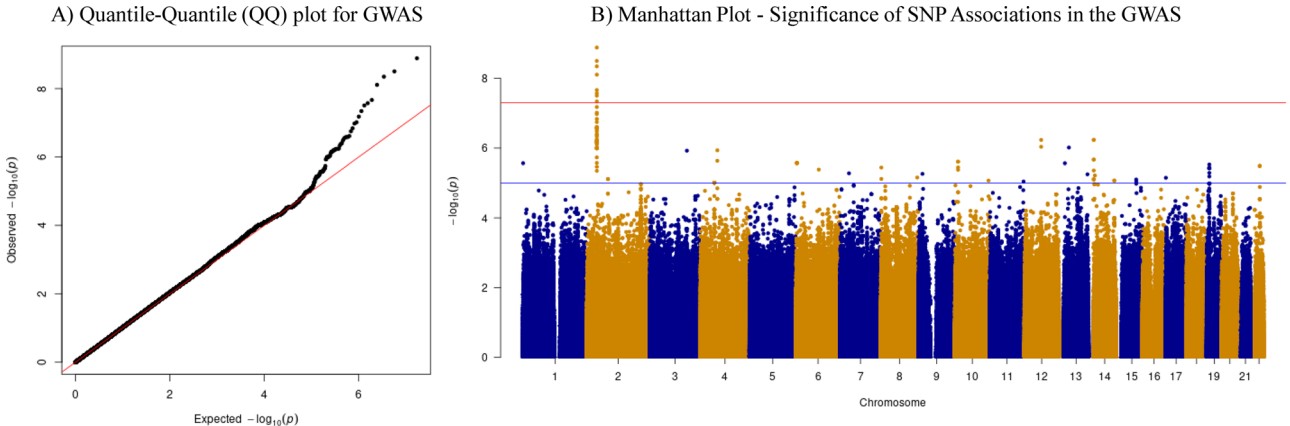

A) Quantile-Quantile (QQ) plot for GWAS

B) Manhattan Plot - Significance of SNP Associations in the GWAS

C) LocusZoom plot of the most strongly associated SNP (rs2113389) in the locus (RMDN2-CYP1B1)

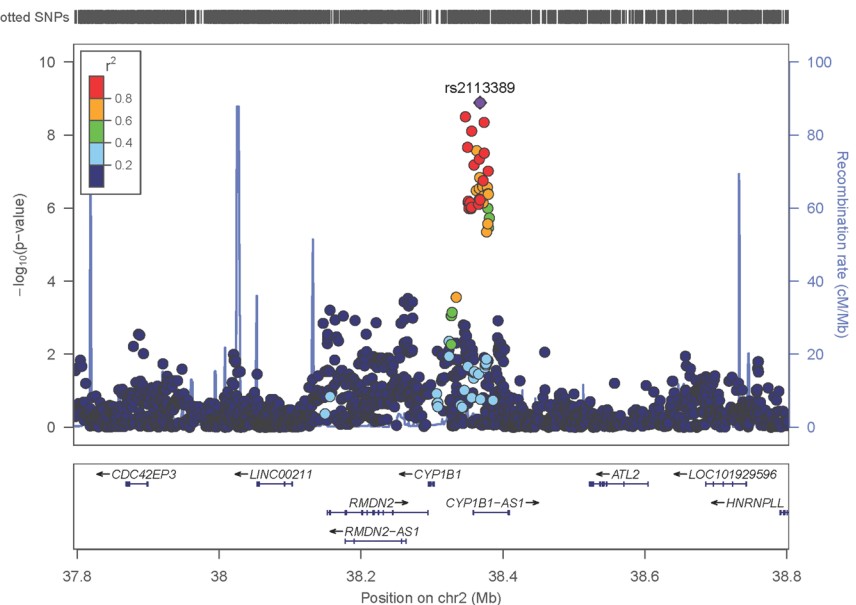

**Fig. 1 | Results of Discovery GWAS for cortical tau deposition.** Quantile-quantile (QQ) (**A**), Manhattan (**B**), and LocusZoom (**C**) plots of genome-wide association study (GWAS) results from seven discovery cohorts ($N = 1446$) using a linear regression model with age, sex, two principal component (PC) factors from population stratification, *APOE4* status, and diagnosis as covariates are shown. The genomic inflation factor is $\lambda = 1.025$ in the Manhattan plot (**B**), the horizontal blue and red lines represent the -$\log_{10}(10^{-5})$ and -$\log_{10}(5.0 \times 10^{-8})$ threshold levels, respectively. Two single nucleotide polymorphisms (SNPs) on chromosome 2 showed highly significant ($< 5.0 \times 10^{-8}$) associations with cerebral tau deposition. The regional association plot (**C**) for the locus that passed genome-wide significance shows the region around the most significant SNP (rs2113389) at the *RMDN2-CYP1B1* locus. SNPs were plotted based on their GWAS $-\log_{10}$ *p*-values and genomic position. The red color scale of $r^2$ values was used to label SNPs based on their degree of linkage disequilibrium with the most significant SNP. Recombination rates calculated from 1000 Genomes Project reference data are also displayed in a blue line corresponding to the right vertical axis. *Note: cerebral tau endophenotype measured as an inverse normal transformed variable of cortical tau SUVR.*

$\lambda = 1.025$; Fig. 1A). We identified a genome-wide significant association of cortical tau with a novel locus at 2p22.2 (Fig. 1B), with two SNPs reaching genome-wide significance (*p*-value $\leq 5 \times 10^{-8}$). The strongest associated SNP is rs2113389, which was directly genotyped. The other SNP (rs918804) is in strong linkage disequilibrium (LD, $r^2 = 0.91$ and $D' = 0.95$). rs2113389 is located on 2p22.2 between *RMDN2*, *CYP1B1*, and non-coding RNA, *CYP1B1-AS1* (Fig. 1C). The minor allele T of rs2113389 (MAF = 0.146) was associated with higher tau (Z score = 5.68; *p*-value = $1.37 \times 10^{-8}$; Heterogeneity $I^2 = 27.8$; Heterogeneity *p*-value = $2.17 \times 10^{-1}$). A replication meta-analysis in five additional cohorts ($n = 1600$) showed that the significant SNPs (rs2113389 and rs918804) in the discovery stage were replicated with the same association direction (Z Score=3.83, *p*-value = $1.26 \times 10^{-4}$, Heterogeneity $I^2 = 52.0$, Heterogeneity *p*-value = $8.02 \times 10^{-2}$; Z-score = $-2.97$, *p*-value = $2.97 \times 10^{-3}$, Heterogeneity $I^2 = 59.5$; Heterogeneity *p*-value = $5.99 \times 10^{-2}$, respectively; Supplementary Fig. 1). -4.3% of the estimated proportional variation in cortical tau in ADNI is explained by rs2113389 and *APOE4* (rs429358).

**Association of rs2113389 genotype with regional and global tau**
Figure 2 shows that both additive (Fig. 2A,B) and dominant models (Fig. 2C,D) demonstrated higher MTL and cortical tau deposition in rs2113389 minor allele (T) carriers. Similar results were observed when stratified by sex (Supplementary Figs. 2,3) and when using SUVR values rather than those with rank-based inverse normal transformation (Supplementary Fig. 4).

**Interaction of rs2113389 genotype with variables of interest**
Main effects of diagnosis and rs2113389 genotype were observed but no interaction effect (Fig. 3A,B). As the pattern of the *RMDN2-CYP1B1* association is similar across diagnoses, this effect is not being fully driven by MCI/AD patients. The effect was similar in both males and

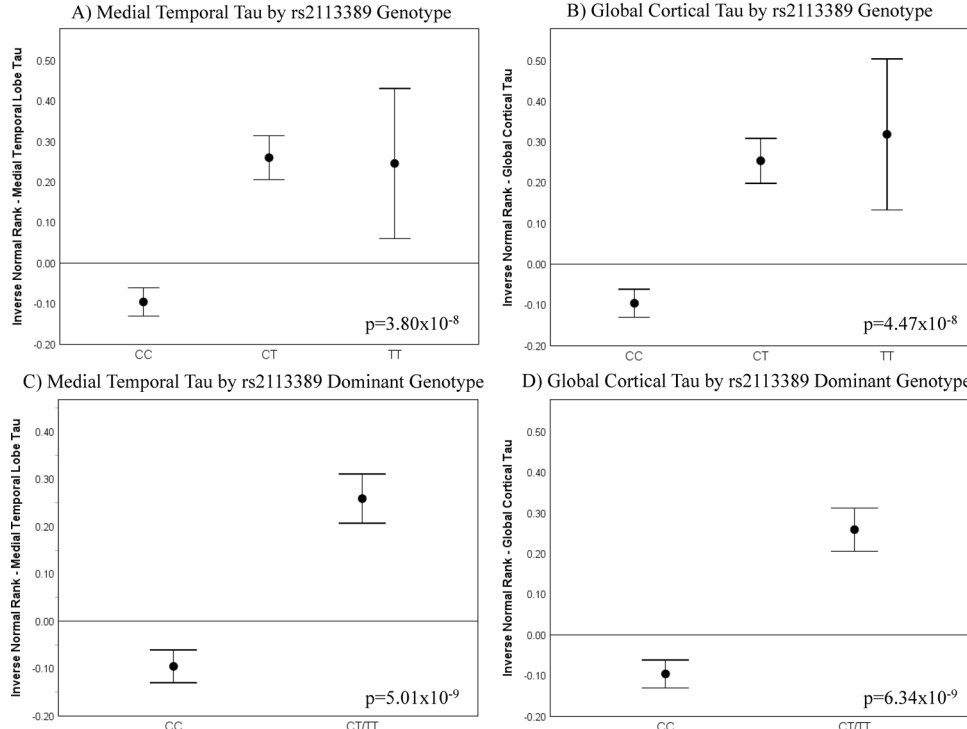

**Fig. 2 | Association of the most significant SNP (rs2113389) at the *RMDN2-CYP1B1* locus with regional and global cortical tau burden.** Using an additive model, the minor allele (T) of rs2113389 is associated with higher tau deposition across participants, with both rs2113389 CT and TT individuals showing significantly greater medial temporal lobe (MTL; **A**) and cortical (**B**) tau deposition than rs2113389 CC individuals. Similar results are seen using a dominant model. Specifically, individuals with one or more minor alleles of rs2113389 show significantly greater tau deposition in the medial temporal lobe (**C**) and cortex (**D**) than rs2113389 CC individuals. One-way ANCOVA models are used with rs2113389 genotype as the independent variable, covaried for age, sex, Aβ positivity, *APOE4* carrier status, and diagnosis. Plots represent mean ± standard error of the mean. Panels include 1,161 individuals (for **A**, **B**, 834 CC, 300 CT, 27 TT); for (**C,D**), 834 CC, 327 CT/TT). Source data are provided as a Source Data file. Aβ amyloid-beta; ANCOVA analysis of covariance; APOE apolipoprotein E; MTL medial temporal lobe; SUVR standardized uptake value ratio. *Note: tau measured as an inverse normal transformed variable of medial temporal and cortical tau SUVR.*

females (Supplementary Fig. 5). Main effects, but no interaction effect, for rs2113389 genotype and *APOE4* were also observed (Fig. 3C,D). The sex-stratified analysis showed similar results in both males and females (Supplementary Fig. 6). Finally, main effects of Aβ positivity and rs2113389 genotype, but no interaction effect were observed (Fig. 3E,F). In the sex-stratified analysis, males and females showed similar results (Supplementary Fig. 7), except for an interaction effect of Aβ positivity and rs2113389 genotype on MTL tau deposition in females (Supplementary Fig. 7C). Similar results were also observed using SUVR values rather than the rank-based inverse normal transformed values (Supplementary Fig. 8).

**Voxel-wise association of rs2113389 genotype with tau**
A voxel-wise analysis of the effect of rs2113389 (voxel-wise $p < 0.05$ (FWE corrected), minimum cluster size (k) = 100 voxels; Fig. 4 and Supplementary Fig. 9) evaluated the topographic pattern of the association. In the dominant model, rs2113389 minor allele carriers (CT or TT; $n = 327$) demonstrated greater tau than rs2113389 CC individuals ($n = 834$; Fig. 4A). Beta-value maps supported the statistical map, showing widespread areas where rs2113389-T carriers show higher tau than non-carriers (Fig. 4B). Using an additive model, rs2113389 CT individuals ($n = 300$) showed higher tau than CC individuals ($n = 834$) in the temporal, parietal, and frontal lobes (Supplementary Fig. 9A), while rs2113389 TT ($n = 27$) showed a focal region of higher frontal tau relative to CC individuals (Supplementary Fig. 9B). Beta-value maps revealed rs2113389 CT individuals showing higher temporal and parietal tau relative to rs2113389 CC individuals (Supplementary Fig. 9C). rs2113389 TT individuals showed widespread higher tau relative to rs2113389 CC individuals, especially in the frontal lobe (Supplementary

Fig. 9D). Finally, the beta-values map shows that rs2113389 TT homozygotes show higher frontal tau than rs2113389 CT heterozygotes (Supplementary Fig. 9E), although this did not reach statistical significance.

**Association of rs2113389 genotype with CSF tau biomarkers**
In addition to the findings with PET, rs2113389 genotype was associated with CSF levels of both total tau and phosphorylated tau 181 (pTau181), with the rs2113389 T-allele associated with higher levels of CSF total tau and pTau181 both in the additive model (Supplementary Fig. 10A,B) and dominant model (Supplementary Fig. 10C,D). We reviewed the GWAS summary statistics from two large-scale GWAS for CSF biomarkers[14,22]. rs1478361 was associated with CSF total-tau levels but not CSF p-Tau levels. rs1478361, which is in strong LD with rs2113389 ($r^2 = 0.96$ and D' = 1.00), was associated with CSF total tau levels ($n = 3,076$; β = 0.0176; $p$-value = 0.0295)[14]. Within the *CYP1B1* locus, the most significant SNPs for CSF p-Tau levels were rs12463523 ($p$-value = 0.0026) from the Deming et al. paper[14] and rs9341266 ($p$-value = 0.0029) from the Jansen et al. paper[22].

**Pathway analysis**
When gene ontology (GO) terms were considered, 480 gene-sets were significant after correction for multiple testing. GO for cell-cell adhesion was the most significant pathway identified (Supplementary Table 13A). GO terms for MHC protein complex, postsynaptic density, regulation of synaptic transmission, and calcium ion transport were also significant. For the KEGG pathway, 44 gene-sets were significant, including cell adhesion molecules, calcium signaling pathways, and axon guidance (Supplementary Table 13B). GO terms for several

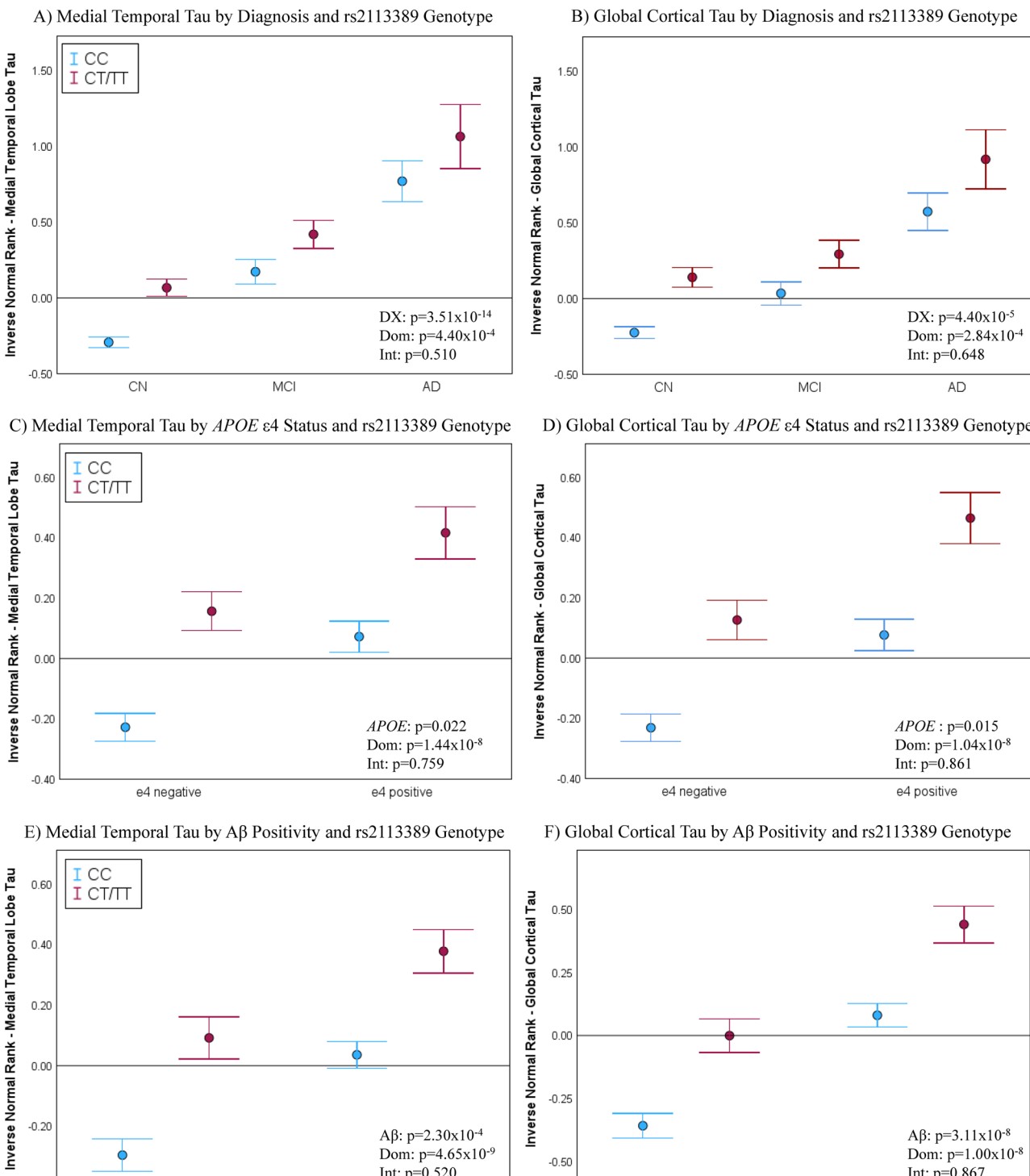

**Fig. 3 | Interaction effect of the most significant SNP (rs2113389) at the *RMDN2-CYP1B1* locus with diagnosis, *APOE ε4* carrier status, and Aβ positivity on regional and cortical tau deposition.** Both diagnosis and rs2113389 dominant genotype are significantly associated with medial temporal lobe (MTL; **A**) and cortical (**B**) tau deposition. *APOE4* carrier status and rs2113389 dominant genotype are significantly associated with MTL (**C**) and cortical (**D**) tau deposition. Significant effects of both Aβ positivity and rs2113389 dominant genotype on MTL (**E**) and cortical (**F**) tau deposition are observed. Two-way ANCOVA models, covaried for age, sex, as well as diagnosis, *APOE4* carrier status, and Aβ positivity where appropriate, are used. Plots are displayed as mean +/−standard error of the mean.

Panels (**A**) and (**B**) include 1161 participants (568 CN-CC, 222 CN-CT/TT, 195 MCI-CC, 75 MCI-CT/TT, 71 AD-CC, 30 AD-CT/TT); panels (**C**) and (**D**) include 1161 participants (468 *APOE4*-/CC, 199 *APOE4*-/CT/TT, 366 *APOE4* + /CC, 128 *APOE4* + /CT/TT); panels (**E**) and (**F**) include 1154 participants (338 Aβ-/CC, 131 Aβ-/CT/TT, 491 Aβ + /CC, 194 Aβ + /CT/TT). Source data are provided as a Source Data file. Aβ amyloid-beta; AD Alzheimer's disease; ANCOVA analysis of covariance; APOE apolipoprotein E; CN cognitively normal; DX diagnosis; Dom rs2113389 dominant genotype (CC vs. CT/TT); Int. interaction; MCI mild cognitive impairment; MTL medial temporal lobe; SUVR standardized uptake value ratio. *Note: tau measured as an inverse normal transformed variable of medial temporal and cortical tau SUVR.*

A) Voxel-wise Statistical Map Showing Significant Regions of Higher Tau in rs2113389-T Carriers Relative to Non-Carriers

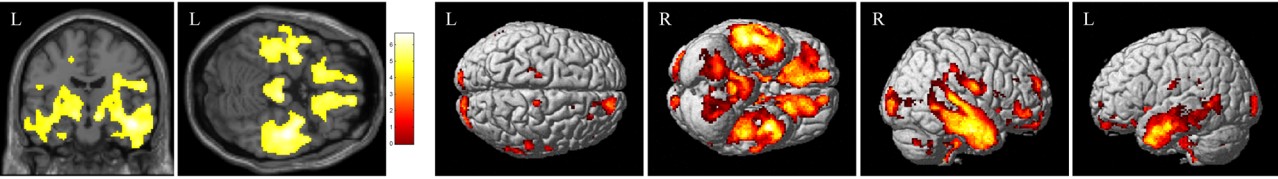

Voxel-wise p<0.05 FWE; Minimum cluster size (k)=100 voxels

B) Voxel-wise Beta Values Map Showing All Regions of Higher Tau in rs2113389-T Carriers Relative to Non-Carriers

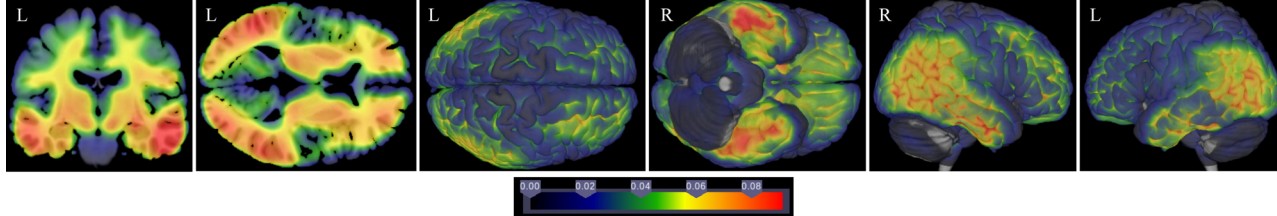

**Fig. 4 | Voxel-wise analysis and visualization of the effect of rs2113389 dominant genotype on tau deposition. A** Widespread regions of association between rs2113389 dominant genotype and tau deposition are observed in the inferior frontal, parietal, and medial and lateral temporal lobes, such that those with one or more minor alleles (T) at rs2113389 show greater tau deposition than CC rs2113389 individuals. Images are displayed at a voxel-wise threshold of $p < 0.05$ with family-wise error correction for multiple comparisons and a minimum cluster size

(k) = 100 voxels. **B** Beta-value maps show widespread regions of higher tau deposition in rs2113389-T carriers relative to non-carriers. Specifically, temporal, parietal, and frontal lobe tau is greater in minor allele carriers than non-carriers. A one-way ANCOVA model is used, covaried for age, sex, diagnosis, APOE4 carrier status, and Aβ positivity. Analyzes include 1154 individuals (829 CC, 325 CT/TT). Aβ amyloid-beta; ANCOVA analysis of covariance; APOE apolipoprotein E.

pathways containing genes near the *CYP1B1* locus were significant, including those that regulate reactive oxygen species, metabolic processes, monooxygenase activity, Golgi organization, and endoplasmic reticulum organization, as well as the KEGG pathway for steroid hormone biosynthesis.

### Gene expression analysis and eQTL analysis

Our genome-wide gene-based association analysis identified two protein coding genes (*CYP1B1* (corrected $p$-value = 0.040)), *RMDN2* (corrected $p$-value = 0.040)), and one non-coding RNA (*CYP1B1-AS1* (corrected $p$-value = 0.040)) associated with tau. Then, our Allen Human Brain Atlas visualization showed that *CYP1B1* was expressed across the whole brain, especially in the insula, orbitofrontal cortex, and temporal lobe. *RMDN2* was also expressed throughout the brain, especially the temporal lobe, visual cortex, frontal and posterior default mode network regions, and sensorimotor cortex (Supplementary Fig. 11). Processed bulk RNA-Seq data from 1917 samples downloaded from the AMP-AD Knowledeg Portal[23–26] was evaluated for these genes. Differential expression of *RMDN2* was seen in the parahippocampal gyrus ($p$-value = 0.004; Fig. 5A), with down-regulation in AD. *CYP1B1* demonstrated differential expression in the temporal cortex ($p$-value = 0.001; Fig. 5B), with upregulation in AD. In eQTL analysis, the rs2113389 was associated with *CYP1B1* expression levels in the temporal cortex, but not with *RMDN2* expression. Specifically, the rs2113389 T-allele was associated with higher temporal *CYP1B1* expression (β = 0.25; $p$-value = 0.02; Fig. 5C). Finally, the rs2113389 T-allele was associated with higher *CYP1B1* expression levels in blood from the eQTLGen consortium database ($n$ = 31,684; Z Score=24.93; $p$-value = 3.6 × 10$^{-137}$).

### Cell type-specific expression and eQTL analysis of CYP1B1

Single-cell expression of *CYP1B1* in ROSMAP single-nucleus RNA-Seq data from the dorsolateral prefrontal cortex downloaded from the AMP-AD Knowledge Portal showed that fibroblasts (Fib) had the highest *CYP1B1* gene expression across all cell types[27]. Among the eight major brain cell types, excitatory neurons (Exc) had the highest *CYP1B1* expression (Fig. 5D). Finally, eQTL analysis of cell type specific *CYP1B1*

expression in excitatory neurons showed that the rs2113389 T-allele was associated with higher cell type-specific *CYP1B1* expression levels ($p$-value = 0.035; Fig. 5E).

### Blood-based DNA methylation QTLs of rs2113389

The DNA methylation QTL (meQTL) analysis of rs2113389 with CpGs in *CYP1B1* in blood identified three CpGs located in the *CYP1B1* gene body[28] associated with rs2113389 ($p$-value < 1 × 10$^{-5}$; Fig. 5F). The rs2113389 T-allele was associated with higher CpG expression levels.

### Cyp1b1 expression and expression changes in the brain of AD mice

*Cyp1b1* expression was increased in the cortex of 6-month-old hTAU mice ($p$-value = 0.038; Fig. 5G). *Cyp1b1* expression also significantly changed with time (genotype*age) in rTg4510 mice (FDR corrected $p$-value = 0.040) but not J20 mice relative to wild-type mice (Fig. 5H, I)[21]. *Cyp1b1* differential expression over time in the TG rTg4510 mice was associated with entorhinal cortex tau pathology (FDR-corrected $p$-value = 0.002; Supplementary Table S5 in Castanho et al.)[21].

## Discussion

We performed a GWAS of cortical tau PET and identified and replicated a novel SNP at the *CYP1B1-RMDN2* locus at 2p22.2. The most significant SNP at the locus was rs2113389, with the minor allele (T) of rs2113389 associated with higher tau across diagnoses. An additive effect of the T-allele with *APOE4* status and Aβ positivity was also observed, with *APOE4*+ and Aβ+ minor T-allele carriers having the highest tau levels. In sex-stratified analyses, generally similar results were observed. Overall, these results provide converging evidence that the minor allele (T) of rs2113389 is a risk variant for high tau. Voxel-wise whole brain analysis confirmed that the rs2113389 T-allele was associated with tau in AD-related cortical regions. These findings also support a previous GWAS of CSF tau, where rs1478361, which is in strong LD with rs2113389 ($r^2$ = 0.96 and D′ = 1.00), was associated with CSF total tau levels ($n$ = 3,076; β = 0.0176; $p$-value = 0.0295)[14]. However, recent large-scale AD GWAS studies have shown that the two SNPs were not significantly associated with AD with different association directions across the

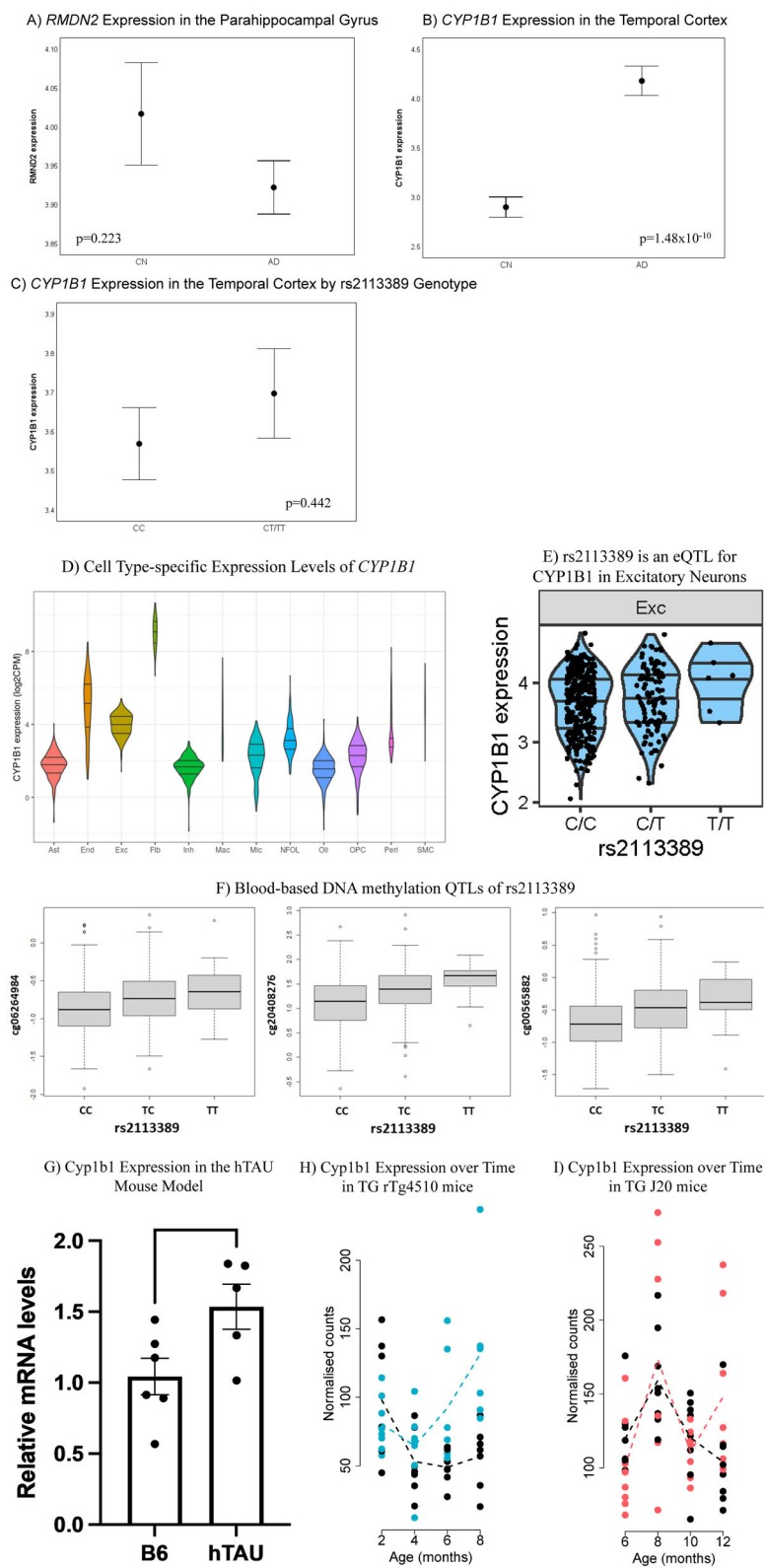

studies (*p*-value > 0.05)[4,29–31]. This lack of significant association may reflect heterogeneity in case-control ascertainment based on clinical diagnosis and is consistent with a selective association elucidated using quantitative endophenotype analysis.

The two protein coding genes at the locus identified in this analysis (*CYP1B1* and *RMDN2*) are highly expressed in the brain in the frontal and temporal lobes (*CYP1B1*) and the cortex (*RMDN2*). Regions showing higher expression levels overlap with the typical patterns of tau deposition, suggesting a spatial relationship between gene expression levels and tau deposition. *RMDN2* (Regulator of Microtubule Dynamics 2) is down-regulated in the parahippocampal gyrus in AD, while *CYP1B1* (Cytochrome P450 Family 1 Subfamily B Member 1) is up-regulated in the temporal cortex in AD. The rs2113389 minor allele is associated with higher temporal cortex *CYP1B1* expression levels.

**Fig. 5 | Gene expression analysis of _RMDN2_ and _CYP1B1_ and expression quantitative trait locus (eQTL) and DNA methylation QTL (meQTL) analysis of rs2113389.** AD patients show downregulated expression of _RMDN2_ in the parahippocampal gyri (**A**) and upregulated expression of _CYP1B1_ in the temporal cortex (**B**) relative to CN using brain tissue-based RNA-Seq data from the AMP-AD project (Panel (**A**), $n = 135$ (26 CN, 109 AD); Panel (**B**), $n = 151$ (71 CN, 80 AD)). **C** In an eQTL analysis, the identified SNP (rs2113389) is associated with _CYP1B1_ expression levels in the temporal cortex ($n = 257$ (188 CC, 69 CT/TT)). One-way ANCOVA models are used in Panels (**A**–**C**), and plots represent the mean ± standard error of the mean. Source data are provided for panels (**A**–**C**) as a Source Data file. Cell type-specific expression levels (**D**) and eQTL in the excitatory neuron (**E**) of _CYP1B1_ gene ($N = 424$) are shown. In (**D**), the x-axis is cell types in ROSMAP DLPFC single-nucleus RNA-Seq data. The y-axis is the $\log_2$ of counts per million mapped reads (CPM) of _CYP1B1_ gene. Expression levels are computed at the donor level by aggregating cells from the same donor. Rare cell types are observed only in a small fraction of donors. Areas of violin plots are scaled to the number of donors. Fibroblasts (Fib) has the highest expression of _CYP1B1_ gene. Among major cell types, excitatory neurons (Exc) has the highest expression. In (**E**), the minor allele (T) of rs2113389 is associated with higher cell type-specific _CYP1B1_ expression levels in the excitatory neuron ($p$-value = 0.035). **F** DNA methylation QTL analysis (_cis_-meQTL) of rs2113389 with CpGs in _CYP1B1_ measured in blood samples from 634 ADNI participants demonstrate three CpGs, located in the _CYP1B1_ gene body region, as significantly associated with rs2113389 ($p$-value = $7.04 \times 10^{-8}$, $5.43 \times 10^{-9}$, and $4.73 \times 10^{-12}$, respectively). **G** _Cyp1b1_ expression (relative mRNA expression levels by qPCR) is increased in the cortex of 6-month-old hTAU mice consistent with our findings in human LOAD ($p$-value = 0.038). The error bars represent the standard error of the mean. **H** _Cyp1b1_ expression (normalized RNA-Seq read counts) significantly changes with time (genotype*age) in TG rTg4510 mice, suggesting _Cyp1b1_ is associated with disease progression in the rTg4510 model. **I** _Cyp1b1_ expression (normalized RNA-Seq read counts) does not change with time (genotype*age) in J20 mice, suggesting that _Cyp1b1_ is not associated with amyloid pathology progression. AD Alzheimer's disease; ADNI Alzheimer's Disease Neuroimaging Initiative; AMP-AD Accelerating Medicines Partnership-AD; ANCOVA analysis of covariance; _cis_-meQTL DNA methylation quantitative trait loci; CN cognitively normal; CpG cytosines followed by guanine residues; CPM counts per million; DLPFC dorsolateral prefrontal cortex; DNA Deoxyribonucleic acid; eQTL expression quantitative trait loci; Exc excitatory neurons; Fib fibroblasts; hTAU humanized tau; ROSMAP Religious Orders Study/Memory and Aging Project; RNA-Seq Ribonucleic acid sequencing; SNP single nucleotide polymorphism.

Fibroblasts and excitatory neurons had the highest expression levels of the _CYP1B1_, and in excitatory neurons, the rs2113389 minor allele was associated with higher _CYP1B1_ expression levels. Blood-based meQTL analysis also supported the impact of rs2113389 on CpGs within the _CYP1B1_ gene, with the rs2113389 T-allele associated with higher CpG expression. Finally, _Cyp1b1_ expression was higher in the cortex of 6-month-old hTAU mice relative to controls. In longitudinal analysis, _Cyp1b1_ expression changed with aging in rTg4510 mice but not J20 mice, suggesting _Cyp1b1_ expression is associated with tau but not amyloid pathology.

_CYP1B1_ is of particular interest as the eQTL analysis shows altered temporal lobe expression in AD patients, and the rs2113389 genotype is linked to temporal lobe _CYP1B1_ expression. _CYP1B1_ is a member of the cytochrome p450 enzyme family (CYP). CYP is present and active in the brain and expressed in a region- and cell-specific manner, including in the blood-brain barrier[32–34]. CYP is responsible for oxidative metabolism of exogenous and endogenous substrates, potentially having both neuroprotective and pathologic roles[33]. CYP is also involved in modulating blood flow, metabolism of fatty acids, cholesterol, and neurotransmitters, and mobilization of intracellular calcium[35–38], suggesting multiple potential roles in AD. Previously, genetic variants in CYP genes have been associated with neurodegenerative diseases, including AD[39,40], as well as Aβ and tau[35,41–43]. _CYP1B1_ regulates endogenous pathways involved in the metabolism of drugs and the synthesis of cholesterols, steroids, and other lipids[44]. While several cytochrome P450 family genes have been implicated in AD, _CYP1B1_ has not previously been directly implicated in AD[39,40,43]. However, _CYP1B1_ may have multiple potential roles related to AD-related tau pathology and has been shown to be a regulator of oxidative stress, which promotes angiogenesis[45,46]. _CYP1B1_ also promotes angiogenesis by suppressing NF-kB activity, which is also implicated in inflammation[47]. Previous studies suggest that _CYP1B1_ inhibition reduced oxidative stress and metabolized cell products that modulate intracellular oxidative stress; however, a lack of _CYP1B1_ leads to increased intracellular oxidative stress in the endothelium[48–50]. _CYP1B1_ may play an important role in high fat diet-associated learning and memory deficits and oxidative damage[50]. Increased brain oxidative stress causes cell damage with aging and is an important pathogenic factor in AD, contributing to tau phosphorylation and the formation of neurofibrillary tangles[51–53]. Functional studies for _RMDN2_ are limited, only showing that it encodes a protein important for regulating microtubule dynamics.

Pathway-based analysis identified enrichment in pathways related to the MHC, postsynaptic membrane, postsynaptic density, synapse organization, and calcium channel activity. MHC pathways have been implicated in large-scale AD genetic associations[4,29,54], along with specific MHC alleles[55]. Microglial activation via MHC class II signaling is increased in regions of phosphorylated tau[56]. Dysfunctional synaptic connections are involved early in AD-related cognitive impairment[57], and tau deposition may induce synaptic impairment and learning deficits[58,59]. Studies also suggest a role for tau at dendritic spines in affecting the trafficking of postsynaptic receptors[60,61]. Finally, the $Ca^{2+}$ signaling and homeostasis are implicated in AD pathology[62] and have been linked to tau phosphorylation[63,64]. Treatments targeting calcium channels are potential pathways for novel therapeutics for neurodegenerative diseases[64].

There are some notable limitations, as this study was primarily observational and composed only of European ancestry cohorts. Multiethnic studies are important, and to be generalizable to other populations, our findings require replication using large community studies or international collaborations. Although similar methodologies were used in all cohorts, subtle differences due to Freesurfer version or slightly different reference regions for SUVR calculation are possible. Further, all cohorts except AIBL-2 employed the same tau PET tracer ([18F]flortaucipir), which may have introduced additional variability. However, the replication of the genetic association in an independent cohort using a different tau PET tracer lends confidence to the generalizability of the findings. Minor sex differences were observed in the pattern of results. Although sex differences are increasingly recognized as important for precision medicine in ADRD, the current study was not designed or powered to thoroughly test these effects. Future studies that assess the presence and pattern of sex differences in longitudinal studies with larger samples are warranted. Even though a number of the cohorts included in the present manuscript have longitudinal follow-up, the current study focused primarily on cross-sectional associations. Future studies to evaluate longitudinal follow-up in these cohorts, including analyses of longitudinal tau PET phenotypes, are also warranted. The Allen Human Brain Atlas results suggest that the genes identified in this analysis are expressed in tau-relevant brain regions. However, these findings do not indicate that expression of these genes is exclusive to brain regions with high tau. Notably, the AHBA did not include patients with ADRD, which limits their utility for disease-related hypotheses. Finally, although we performed the largest GWAS of tau PET to date, our meta-analysis had limited statistical power due to the moderate sample size for genetic association. Additional independent large cohorts with tau PET and GWAS data are needed.

In summary, GWAS of tau PET identified novel genetic variants in a locus (*CYP1B1-RMDN2*) that influences MTL and cortical tau levels. The mechanistic significance of this locus was supported by a range of independent functional genomic observations in humans and model systems. Taken together, these results can inform future biomarker and therapeutic development.

## Methods

### Participants

The study complies with all relevant ethical regulations. Informed consent was obtained for all participants according to the Declaration of Helsinki, and studies were approved by the Human Subjects & Institutional Review Boards (IRB) at Indiana University (Alzheimer's Disease Genomics: Systems Biology and Endophenotypes, 1806870105) as well as the Institutional Review Boards at each participating site. All animal studies were performed in accordance with US National Institutes of Health guidelines on animal care and were approved by appropriate Institutional Animal Care and Use Committees[21]. Descriptions of all cohorts are found in the Supplementary information (Supplementary Tables 1–12). Participants were from the Alzheimer's Disease Neuroimaging Initiative (ADNI; http://adni.loni.usc.edu), ADNI-Department of Defense (ADNI-DoD), Indiana Memory and Aging Study (IMAS), Avid A05 clinical trial (A05), Anti-Amyloid Treatment in Asymptomatic Alzheimer's (A4) and Longitudinal Evaluation of Amyloid Risk and Neurodegeneration (LEARN) studies, Harvard Aging Brain Study (HABS), University of Pittsburgh Alzheimer's Disease Research Center (UPitt ADRC), Mayo Clinic Study of Aging (MCSA), Memory and Aging Project (MAP) at the Knight Alzheimer's Disease Research Center (Knight-ADRC), the Australian Imaging, Biomarker and Lifestyle Study (AIBL; https://aibl.org.au/), and the Berkeley Aging Cohort Study (BACS). The discovery sample included ADNI, ADNI-DoD, IMAS, A05, A4, HABS, UPitt ADRC. The replication sample included MCSA, MAP-Knight ADRC, AIBL, and BACS. Post-hoc analyzes of interactions with diagnosis, APOE4, and Aβ positivity, and voxel-wise analyzes were performed in 1161 individuals from ADNI, ADNI-DoD, IMAS, A05, A4, and LEARN.

### Genotyping and imputation

Participants were genotyped using several genotyping platforms. Ungenotyped SNPs were imputed separately in each cohort using the Haplotype Reference Consortium (HRC) data as a reference panel[65]. Before imputation, standard sample and SNP quality control (QC) procedures were performed[66]. Only non-Hispanic participants of European ancestry by multidimensional scaling analysis were selected[67]. Imputation and QC procedures were performed as described previously[68].

### Statistical analysis

**Genome-wide association analysis (GWAS).** Cortical tau deposition (the weighted average SUVR of all cortical regions from FreeSurfer version 6.1 parcellation (aparc)) followed a normal distribution after a rank-based inverse normal transformation. Using imputed genotypes, a GWAS of cortical tau was performed using a linear regression model with age, sex, two principal component (PC) factors from population stratification, APOE4 status, and diagnosis as covariates using PLINK[69]. APOE4 status was included as a covariate because its effect was modeled to understand the contribution of the discovered *CYP1B1-RMDN2* locus above and beyond *APOE4* and to assess whether there is epistasis with *APOE4*. A fixed effect meta-analysis with an inverse variance weighted approach was performed using METAL, and a heterogeneity analysis in METAL was performed to evaluate the possible effect of study heterogeneity on the results[31,54,70]. See Supplementary information for more details. The proportion of variance in tau explained was assessed using the Genome-wide Complex Trait Analysis (GCTA) tool[71].

**Gene-set enrichment analysis.** Gene-set enrichment analysis was performed using GWAS summary statistics to identify pathways and functional gene sets associated with cortical tau deposition using the GSA-SNP software[72], as described in the Supplementary information.

**Gene-based association analysis.** Genome-wide gene-based association analysis was performed using GWAS p-values and the KGG software as described previously[73,74] and in the Supplementary information.

**Interaction with diagnosis, APOE genotype, and Aβ positivity.** The effect of the top identified SNP (rs2113389 – dominant model) and its interaction with diagnosis, APOE4 status, and Aβ positivity, on global and medial temporal lobe (MTL) tau was assessed. Differential effects by sex were also evaluated using stratified analysis. See methods in Supplementary information.

**Detailed whole-brain imaging analysis.** Tau PET SUVR images ($n = 1161$) were used in a voxel-wise statistical analysis of the effect of the top identified SNP on tau using SPM12 (www.fil.ion.ucl.ac.uk/spm/) in a post-hoc analysis (described in the Supplementary information).

**CSF tau analysis.** CSF total tau and phosphorylated tau 181 values from the Roche Elecsys assay[75,76] were available for a subset ($n = 525$; 332 CN, 153 MCI, 40 AD) of the ADNI and ADNI-DoD cohorts. Total tau and pTau181 levels were not normally distributed, and thus, we transformed using a natural log before analysis. A one-way ANOVA with rs2113389 genotype as the independent variable using both an additive model and dominant model was used to test the association of rs2113389 genotype and CSF total tau and pTau181 levels, covaried for age, sex, *APOE ε4* carrier status, and diagnosis.

**AMP-AD bulk RNA-Seq data in the post-mortem human brain.** Processed RNA-Seq data from seven brain regions in three cohorts were downloaded from the AMP-AD Knowledge Portal (https://doi.org/10.7303/syn2580853) and analyzed as discussed in the Supplementary information[26]. The eQTLGen[77] consortium database ($n = 31,684$) was used for eQTL of rs2113389 with CYP1B1 expression in blood.

**Single-nucleus RNA-Seq (snRNA-Seq) preprocessing and analysis.** Processed snRNA-Seq data from frozen brain tissue specimens ($n = 479$) from the dorsolateral prefrontal cortex in the Religious Orders Study/Memory and Aging Project (ROSMAP) was downloaded from the AMP-AD Knowledge Portal (https://www.synapse.org/#!Synapse:syn31512863)[27,78].

**Allen Human Brain Atlas data and analysis.** Regional gene expression profiles for CYP1B1 and RMDN2 were downloaded from brain-wide microarray-based transcriptome data from the Allen Human Brain Atlas (https://human.brain-map.org/microarray/search), as described in the Supplementary information[79,80].

**ADNI DNA methylation data.** ADNI DNA methylation data was downloaded from the ADNI LONI database (https://adni.loni.usc.edu/), where Illumina EPIC chips (Illumina, Inc., San Diego, CA, USA) were used to profile DNA methylation in 1920 blood or buffy coats samples including 200 duplicate samples according to the Illumina protocols[28]. A detailed protocol has been published previously[28,81,82], and further methods are described in the Supplementary information.

**AD pathology mouse model analysis.** hTau mouse model: Generation of the hTAU mice, as well as brain extraction and tissue processing, was described previously[19,20,83,84] and in the Supplementary information. Student's t-test was performed for qPCR results comparing C57BL/6J (B6; wild type) and hTAU mice. rTg4510 and J20

**mouse model:** Mice harboring human tau (rTg4510) and amyloid precursor protein (J20) mutations were used to investigate gene expression changes of the top identified gene[21]. The rTg4510 and J20 mouse models and experimental models and methods were described previously[21,85–88], and are briefly summarized, along with statistical methods used, in the Supplementary information.

## Reporting summary

Further information on research design is available in the Nature Portfolio Reporting Summary linked to this article.

## Data availability

Summary statistics for the discovery analysis are available in the Alzheimer's Disease Neuroimaging Initiative Laboratory of NeuroImaging repository (ADNI LONI; https://ida.loni.usc.edu/pages/access/studyData.jsp?categoryId=18&subCategoryId=28). Referenced data, including imaging, cognitive, clinical and genetic data from ADNI, A4, and ADNI-DoD can be requested through the Laboratory of NeuroImaging (LONI; https://www.loni.usc.edu/). Imaging data for AIBL is available through the Laboratory of NeuroImaging (LONI; https://www.loni.usc.edu/), while genetic and other data is available by request from the study PIs. Referenced imaging, cognitive, clinical, and genetic data from the other human cohorts (IMAS, A05, HABS, UPitt ADRC, BCSA, MCSA, and the Knight ADRC) is not publicly available and must be requested directly from the study PIs. ADNI DNA methylation data was downloaded from the ADNI LONI database (https://adni.loni.usc.edu/)[28]. Brain-wide microarray-based transcriptome data from the Allen Human Brain Atlas is available through the Allen Brain Map portal (https://human.brain-map.org/microarray/search). RNA-Seq data is available through the AMP-AD Knowledge Portal (https://doi.org/10.7303/syn2580853)[26]. ROSMAP single-nucleus RNA-Seq data is available through the AD Knowledge Portal (https://www.synapse.org/#!Synapse:syn31512863)[27]. Data is available for general research use according to the following requirements for data access and data attribution (https://adknowledgeportal.synapse.org/DataAccess/Instructions). The data from the rTg4510 and J20 mouse models is available in a previous paper[21]. The data from the hTau mouse model is provided. No primary data was generated in this study, as all data used were reference datasets. Source data for the figures included in this paper are provided. Source data are provided with this paper.

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

## Acknowledgements

Drs. Kwangsik Nho and Shannon Risacher had full access to all the data in the study and take responsibility for the integrity of the data and the accuracy of the data analysis. Data collection and sharing for this project was funded by the Alzheimer's Disease Neuroimaging Initiative (ADNI) (National Institutes of Health Grant U01AG024904) (MWW) and DOD ADNI (Department of Defense award number W81XWH-12-2-0012) (MWW). ADNI is funded by the National Institute on Aging, the National Institute of Biomedical Imaging and Bioengineering, and through generous contributions from the following: AbbVie, Alzheimer's Association; Alzheimer's Drug Discovery Foundation; Araclon Biotech; BioClinica, Inc.; Biogen; Bristol-Myers Squibb Company; CereSpir, Inc.; Cogstate; Eisai Inc.; Elan Pharmaceuticals, Inc.; Eli Lilly and Company; EuroImmun; F. Hoffmann-La Roche Ltd and its affiliated company Genentech, Inc.; Fujirebio; GE Healthcare; IXICO Ltd.; Janssen Alzheimer Immunotherapy Research & Development, LLC.; Johnson & Johnson Pharmaceutical Research & Development LLC.; Lumosity; Lundbeck; Merck & Co., Inc.; Meso Scale Diagnostics, LLC.; NeuroRx Research; Neurotrack Technologies; Novartis Pharmaceuticals Corporation; Pfizer Inc.; Piramal Imaging; Servier; Takeda Pharmaceutical Company; and Transition Therapeutics. The Canadian Institutes of Health Research is providing funds to support ADNI clinical sites in Canada. Private sector contributions are facilitated by the Foundation for the National Institutes of Health (http://www.fnih.org). The grantee organization is the Northern California Institute for Research and Education, and the study is coordinated by the Alzheimer's Therapeutic Research Institute at the University of Southern California. ADNI data are disseminated by the Laboratory for Neuro Imaging at the University of Southern California. AVID Radiopharmaceuticals, Inc., supplied the AV-1451 precursor, chemistry production advice and oversight; the FDA provided regulatory cross-filing permission and documentation needed for work in the MCSA cohort. AD Knowledge Portal: AMP-AD datasets: The results published here are in whole or in part based on data obtained from the AMP-AD Knowledge Portal (https://doi.org/10.7303/syn2580853).
**Mayo Clinic:** The Mayo RNAseq study data was led by Dr. Nilüfer Ertekin-Taner, Mayo Clinic, Jacksonville, FL as part of the multi-PI U01 AG046139 (MPI: NET, Todd E. Golde, Cory Funk). Samples were provided from the following sources: The Mayo Clinic Brain Bank and Banner Sun Health Research Institute. Data collection was supported through funding by NIA grants P50 AG016574 (CRJ), R01 AG032990 (NET), U01 AG046139 (NET), R01 AG018023 (Steven G. Younkin), U01 AG006576 (RCP, NET), U01 AG006786 (RCP, CRJ, PV), R01 AG025711 (Dennis W. Dickson), P01 AG017216 (Dennis W. Dickson), Karen Duff, Matthew F. Farrer, John A. Hardy, Michael L. Hutton, Shu-Hui C. Yen), R01 AG003949 (Carol A. Derby), NINDS grant R01 NS080820 (NET), CurePSP Foundation, and support from Mayo Foundation. Study data includes samples collected through the Sun Health Research Institute Brain and Body Donation Program of Sun City, Arizona. The Brain and Body Donation Program is supported by the National Institute of Neurological Disorders and Stroke (U24 NS072026 (Thomas G. Beach), National Brain and Tissue Resource for Parkinsons Disease and Related Disorders), the National Institute on Aging (P30 AG19610 (Thomas G. Beach), Arizona Alzheimers Disease Core Center), the Arizona Department of Health Services (contract 211002, Arizona Alzheimers Research Center), the Arizona Biomedical Research Commission (contracts 4001, 0011, 05-901 and 1001 to the Arizona Parkinson's Disease Consortium) and the Michael J. Fox Foundation for Parkinsons Research. **MSBB:** These data were generated from postmortem brain tissue collected through the Mount Sinai VA Medical Center Brain Bank and were provided by Dr. Eric Schadt from Mount Sinai School of Medicine. **ROSMAP:** Study data were provided by the Rush Alzheimer's Disease Center, Rush University Medical Center, Chicago. Data collection was supported through funding by NIA grants P30AG10161 (ROS), R01AG15819 (ROSMAP; genomics and RNAseq), R01AG17917 (MAP), R01AG30146 (DAB), R01AG36042 (DAB) (5hC methylation, ATACseq), RC2AG036547 (H3K9Ac), R01AG36836 (DAB) (RNAseq), R01AG48015 (DAB) (monocyte RNAseq) RF1AG57473 (DAB) (single nucleus RNAseq), U01AG32984 (DAB) (genomic and whole exome sequencing), U01AG46152 (DAB) (ROSMAP AMP-AD, targeted proteomics), U01AG46161 (DAB) (TMT proteomics), U01AG61356 (DAB) (whole genome sequencing, targeted proteomics, ROSMAP AMP-AD), the Illinois Department of Public Health (ROSMAP), and the Translational Genomics Research Institute (genomic). Additional phenotypic data can be requested at www.radc.rush.edu. Data collection and sharing for this project was funded by the Alzheimer's Disease Neuroimaging Initiative (ADNI) (National Institutes of Health Grant U19 AG024904, PSA) and DOD ADNI (Department of Defense award number W81XWH-12-2-0012, MWW). Additional support for data collection and/or analysis was provided by R01 LM012535 (KN), R03 AG054936 (KN), P30 AG010133 (LGA), P30 AG072976 (LGA), R01 AG019771 (AJS), R01 AG057739 (Louise Brea Perry, AJS, LA), R01 LM013463 (Li Shen, Jason H. Moore, AJS), R01 AG068193 (Jeanne Mandelblatt, AJS), U01 AG068057 (Paul M. Thompson, Christos Davatzikos, Heng Huang, AJS, Li Shen), U01 AG072177

(AJS, Dong Young Lee, KN), R01 LM011360 (Li Shen, Jason H. Moore, AJS), DOD W81XWH-14-2-0151 (Thomas W. McAllister), NIGMS P50GM115318 (Ronald M. Krause, Aldons Jake Lusis, Marisa Wong Medina, Carlos Iribrren), NCATS UL1 TR001108 (Anantha Shekar), K01 AG049050 (SLR), R01 AG061788 (SLR), R01 AG052446 (ADC), R01 AG052521 (Beth Snitz), RF1 AG052525 (OLL), P30 AG066468 (ADC, MIK, OLL), P01 AG025204 (ADC, VLV, OLL), and Donor's Cure Foundation. The AIBL study has received partial financial support from the Alzheimer's Association (US), the Alzheimer's Drug Discovery Foundation, an Anonymous foundation (a philanthropic foundation based in the US; one of the conditions of the funding is maintenance of anonymity), the Science and Industry Endowment Fund, the Dementia Collaborative Research Centres, the Victorian Government's Operational Infrastructure Support program, the Australian Alzheimer's Research Foundation, the National Health and Medical Research Council (NHMRC) Australia, and The Yulgilbar Foundation. Numerous commercial interactions have supported data collection and analyzes. In-kind support has also been provided by Sir Charles Gairdner Hospital, Cogstate Ltd., Hollywood Private Hospital, The University of Melbourne, and St Vincent's Hospital. Additional support for data collection and/or analysis was provided by NHMRC grants (GNT1161706, GNT1191535) awarded to SML. Support for the Harvard Aging Brain Study (HABS) was provided by NIH-NIA Program Project P01-AG036694 (KJ). NIH-NIA K23AG062750 (HSY). The A4 study was funded by the National Institute on Aging (grants U19AG010483 (PSA) and R01AG063689 (RAS), Eli Lilly and Co, and several philanthropic organizations (NCT02008357, RS, KJ, PA). The Mayo Clinic Study of Aging (MCSA) funding includes NIH grants U01 AG006786 (RCP), R01 NS097495 (PV), R01 AG56366 (RCP, CRJ, Michelle M. Mielke), P50 AG016574 (CRJ, NET, RCP, PV), P30 AG062677 (RCP), R37 AG011378 (CRJ), R01 AG041851 (CRJ), R01 AG034676 (Alanna M. Chamberlain, Hilal Maradit Kremers, Walter A. Rocca Barbara P. Yawn Jennifer Lynn St. Sauver, Veronique Lee Roger, Nathan K. Lebrasseur, Janet E. Olson), C06 RR018898 (John C. Burnett). The GHR Foundation, the Alexander Family Alzheimer's Disease Research Professorship of the Mayo Clinic, the Alzheimer's Association, the Mayo Foundation for Medical Education and Research, the Liston Award, the Elsie and Marvin Dekelboum Family Foundation, the Schuler Foundation. National Centralized Repository for Alzheimer's Disease and Related Dementias (NCRAD): U24AG021886 (TF). This work was supported by grants from the National Institutes of Health, R01AG044546 (CC), P01AG003991 (TB), RF1AG053303 (CC), RF1AG058501 (CC), U01AG058922 (CC), and the Chan Zuckerberg Initiative (CZI), the Michael J. Fox Foundation, the Department of Defense (LI- W81XWH2010849 (CC, TJH, MA) and the Alzheimer's Association Zenith Fellows Award (ZEN-22-848604, CC). The recruitment and clinical characterization of research participants at Washington University were supported by NIH P30AG066444 (CC), P01AG03991 (David A. Balota), and P01AG026276 (TB). This work was supported by access to equipment made possible by the Hope Center for Neurological Disorders, the Neurogenomics and Informatics Center (NGI: https://neurogenomics.wustl.edu/) and the Departments of Neurology and Psychiatry at Washington University School of Medicine. Additional funding from NIH National Institute on Aging includes R01 AG059716 (TJH), R01 AG061518 (TJH), R01 AG034570 (WJJ), and P30 AG066468 (ADC, MIK, OLL). The University of Pittsburgh School of Public Health, Department of Genetics includes R01 AG064877 (MIK, CC) and P30 AG066468 (ADC, MIK, OLL). California- Lawrence Berkeley National Laboratory includes R01 AG034570 (WJJ) and R01 AG062542 (WJJ). The rTg4510 and J20 mouse work was funded in part through the Medical Research Council (MRC) Proximity to Discovery: Industry Engagement Fund (Precision Medicine Exeter Innovation Platform reference MC_PC_14127, IC), an MRC Clinical Infrastructure award (MR/ M008924/1, JM), a Wellcome Trust Multi-User Equipment Award (WT101650MA, IC, JM) and through a research grant from Alzheimer's Research UK (ARUK-PG2018B-016, JM). Additional support for these studieswas provided by the NINDS grant R01-NS080820 (NET), NIA grant R01-AG061796 (NET), NIA grant U19-AG074879 (NET, AS, KN), Alzheimer's Association Zenith Fellows Award (NET).

## Author contributions

K.N., S.L.R., and A.J.S. were involved with study design, statistical analysis, data generation, and drafting of the final manuscript. P.J.B. was involved with drafting of the final manuscript. L.G.A., J.B., M.R.F., T.F., K.N., P.A., R.S., B.H., S.S., D.S., C.R.J., W.J.J., S.L., A.V., J.F.W., V.D., S.M.L., T.P., C.C.R., V.L.V., L.D., T.J.H., J.L., E.M., R.F.B., K.J., H.S.Y., R.C.P., V.K.R., P.V., A.D.C., K.H.F., M.I.K., O.L.L., D.A.B., M.A., T.B., C.C., D.H., P.L.D.J., M.F., V.J., B.T.L., A.P.T., I.C., J.M., M.W.W., C.L.M. and N.E.T. were all involved with generation of the imaging, genetic, other "omics," and/or animal model data. R.D., K.F., T.Y.J., J.P.K., K.N., B.H., A.V., V.D., V.L.V., S.M.L., T.P., L.D., T.J.H., J.L., E.M., R.F.B., H.S.Y., A.D.C., K.H.F., D.A.B., V.J., A.P.T., S.G., M.Y., T.R. and I.C. were involved with processing and analysis of imaging and "omics" data. All authors reviewed and approved the final submitted manuscript.

## Competing interests

Dr. Apostolova received grant or other financial support from the National Institutes of Health (NIH), Alzheimer's Association, AVID Pharmaceuticals, Life Molecular Imaging, Roche Diagnostics, and Eli Lilly. In addition, she has received consulting fees from Biogen, Two Labs, IQIVA, Florida Department of Health, Genentech, NIH Biobank, Eli Lilly, GE Healthcare, Eisai, and Roche Diagnostics. She has also received payment or honoraria from American Academy of Neurology, MillerMed, National Alzheimer's Coordinating Center CME, CME Institute, APhA, Purdue University, Mayo Clinic, MJH Physician Education Resource, and Ohio State University. She received support for travel from the Alzheimer's Association. She has served on Data Safety and Monitoring or Advisory Boards for IQVIA, UAB Nathan Schock Center, New Mexico Exploratory ADRC, and NIA R01 AG061111. She has a leadership role in multiple committees, including the Medical Science Council of the Alzheimer's Association Greater Indiana Chapter, the Alzheimer's Association Science Program Committee, and the FDA PCNS Advisory Committee. Finally, Dr. Apostolova holds stock in Cassava Neurosciences and Golden Seeds. Dr. Foroud receives support from multiple NIH grants (U24 NS095871, U24 AG021886, U24 AG056270, U01 AA026103, U10 AA008401, P30 AG010133, R01 AG019771, U01 AG032984, P30 AR072581, U01 AG057195, UL1 TR002529, U19 AG063911; U19 AG063744, U19 AG068054, R01 AG069453, U54 CA196519, R01 AG061146, R01 AG073267, R01 AG074971, U19 AG071754, R01 AG055444, R01 AG070349, U19 AG024904, R01 AG076634, U19 AG079774, U54 CA280897, U19 NS120384); the Michael J. Fox Foundation (MJFF001948); Cohen Veterans Biosciences; The Parkinson's Disease Foundation; Children's Tumor Foundation; Broad Institute; Lumind Foundation; and Gates Venture (0432-06-120975). Dr. Jagust has served as a consultant for Biogen, Eisai, Lilly, and Bioclinica. He has an equity interest in Optoceuticals. Aparna Vasanthakumar and Jeffrey F. Waring are employees of AbbVie and may own AbbVie stock. Dr. Hohman receives support from multiple NIH grants (U24-AG074855, P20-AG068082, R01-AG061518, R01-AG059716, R01-AG074012, RF1-AG059869). He also sits on the advisory board for Vivid Genomics and is a Senior Associate Editor for Alzheimer's and Dementia: Translational Research and Clinical Intervention. Hyun-Sik Yang received personal fees (honorarium) from Genentech, Inc outside the submitted work. Dr. Vemuri receives funding support from the NIH. Dr. Cruchaga has received research support from: GSK and EISAI. The funders of the study had no role in the collection, analysis, or interpretation of data; in the writing of the report; or in the decision to submit the paper for publication. Dr. Cruchaga is a member of the advisory board of Vivid Genomics and Circular Genomics. Dr. Saykin receives support from multiple NIH grants (P30 AG010133, P30 AG072976, R01 AG019771, R01 AG057739, U19 AG024904, R01 LM013463, R01 AG068193, T32 AG071444, U01 AG068057, U01 AG072177 and U19 AG074879). He has also received

support from Avid Radiopharmaceuticals, a subsidiary of Eli Lilly (in kind contribution of PET tracer precursor); Bayer Oncology (Scientific Advisory Board); Eisai (Scientific Advisory Board); Siemens Medical Solutions USA, Inc. (Dementia Advisory Board); NIH NHLBI (MESA Observational Study Monitoring Board); Springer-Nature Publishing (Editorial Office Support as Editor-in-Chief, Brain Imaging and Behavior). Dr. Ertekin-Taner receives support from multiple NIH grants (R01 AG061796, U19 AG074879, U01 AG046139, KL2 TR002379), Alzheimer's Association Zenith Fellows Award and Florida Department of Health. She serves as an advisor to the Framingham Heart Study (NIH NHLBI 75N92019D00031/ 75N92019F00125). She also has a provisional patent application unrelated to this work. The remaining authors declare no competing interests.

## Additional information

Kwangsik Nho[1,2,3,4,41], Shannon L. Risacher [1,3,41], Liana G. Apostolova[1,3,5,6], Paula J. Bice [1,3], Jared R. Brosch [3,5], Rachael Deardorff[3,5], Kelley Faber [6,7], Martin R. Farlow [3,5], Tatiana Foroud [3,6,7], Sujuan Gao[3,8], Thea Rosewood [1,2,3], Jun Pyo Kim [1,2,3], Kelly Nudelman [3,6,7], Meichen Yu [1,3], Paul Aisen[9], Reisa Sperling[10], Basavaraj Hooli[11], Sergey Shcherbinin[11], Diana Svaldi[11], Clifford R. Jack Jr. [12], William J. Jagust [13], Susan Landau[13], Aparna Vasanthakumar [14], Jeffrey F. Waring[14], Vincent Doré [15,16], Simon M. Laws [17], Colin L. Masters[18], Tenielle Porter [17], Christopher C. Rowe [16,18], Victor L. Villemagne[16,19], Logan Dumitrescu[20,21], Timothy J. Hohman [20,21], Julia B. Libby[20], Elizabeth Mormino[22], Rachel F. Buckley [10], Keith Johnson[10,23], Hyun-Sik Yang [10,24], Ronald C. Petersen [25], Vijay K. Ramanan [25], Nilüfer Ertekin-Taner [26,27], Prashanthi Vemuri [12], Ann D. Cohen[19], Kang-Hsien Fan[28], M. Ilyas Kamboh [28], Oscar L. Lopez [19,29], David A. Bennett[30], Muhammad Ali [31], Tammie Benzinger[32], Carlos Cruchaga [31,33], Diana Hobbs [32], Philip L. De Jager [34], Masashi Fujita[34], Vaishnavi Jadhav [6,35], Bruce T. Lamb [3,6,35], Andy P. Tsai [22,35,36], Isabel Castanho [37,38], Jonathan Mill [37], Michael W. Weiner[39,40], for the **Alzheimer's Disease Neuroimaging Initiative (ADNI)\*, the Department of Defense Alzheimer's Disease Neuroimaging Initiative (DoD-ADNI)\*, the Anti-Amyloid Treatment in Asymptomatic Alzheimer's Study (A4 Study) and Longitudinal Evaluation of Amyloid Risk and Neurodegeneration (LEARN)\*, the Australian Imaging, Biomarker & Lifestyle Study (AIBL)\*,** Andrew J. Saykin [1,2,3,5,6] ✉

[1]Center for Neuroimaging, Department of Radiology and Imaging Sciences, Indiana University School of Medicine, Indianapolis, USA. [2]Center for Computational Biology and Bioinformatics, Indiana University School of Medicine, Indianapolis, USA. [3]Indiana Alzheimer's Disease Research Center, Indiana University School of Medicine, Indianapolis, USA. [4]Department of BioHealth Informatics, Indiana University, Indianapolis, USA. [5]Department of Neurology, Indiana University School of Medicine, Indianapolis, USA. [6]Department of Medical and Molecular Genetics, Indiana University School of Medicine, Indianapolis, USA. [7]National Centralized Repository for Alzheimer's Disease and Related Dementias, Indiana University School of Medicine, Indianapolis, USA. [8]Department of Biostatistics, Indiana University School of Medicine, Indianapolis, USA. [9]Department of Neurology, Keck School of Medicine, University of Southern California, San Diego, USA. [10]Department of Neurology, Massachusetts General Hospital, Harvard Medical School, Boston, USA. [11]Eli Lilly and Company, Indianapolis, USA. [12]Department of Radiology, Mayo Clinic, Rochester, USA. [13]UC Berkeley Helen Wills Neuroscience Institute, University of California - Berkeley, Berkeley, USA. [14]Genomics Research Center, AbbVie, North Chicago, USA. [15]CSIRO Health and Biosecurity, Melbourne, Australia. [16]Department of Molecular Imaging & Therapy, Austin Health, Heidelberg, Australia. [17]Centre for Precision Health, School of Medical and Health Sciences, Edith Cowan University, Joondalup, Australia. [18]Florey Institute of Neuroscience and Mental Health and The University of Melbourne, Parkville, Australia. [19]Department of Psychiatry, University of Pittsburgh School of Medicine, Pittsburgh, USA. [20]Vanderbilt Memory & Alzheimer's Center, Vanderbilt University Medical Center, Nashville, USA. [21]Vanderbilt Genetics Institute, Vanderbilt University Medical Center, Nashville, USA. [22]Department of Neurology & Neurological Sciences, Stanford University, Stanford, USA. [23]Department of Radiology, Massachusetts General Hospital, Harvard Medical School, Boston, USA. [24]Center for Alzheimer's Research and Treatment, Department of Neurology, Brigham and Women's Hospital, Harvard Medical School, Boston, USA. [25]Department of Neurology, Mayo Clinic, Rochester, USA. [26]Department of Neurology, Mayo Clinic, Jacksonville, USA. [27]Department of Neuroscience, Mayo Clinic, Jacksonville, USA. [28]Department of Human Genetics, University of Pittsburgh, Pittsburgh, USA. [29]Department of Neurology, University of Pittsburgh School of Medicine, Pittsburgh, USA. [30]Department of Neurological Sciences, Rush Medical College, Rush University, Chicago, USA. [31]Department of

Psychiatry, Washington University, St. Louis, USA. [32]Department of Radiology, Washington University School of Medicine, St. Louis, USA. [33]NeuroGenomics and Informatics Center, Washington University School of Medicine, St. Louis, USA. [34]Center for Translational and Computational Neuroimmunology, Department of Neurology and Taub Institute for Research on Alzheimer's Disease and the Aging Brain, Columbia University Irving Medical Center, New York, USA. [35]Stark Neuroscience Research Institute, Indiana University School of Medicine, Indianapolis, USA. [36]Wu Tsai Neurosciences Institute, Stanford University School of Medicine, Stanford, USA. [37]Department for Clinical and Biomedical Sciences, University of Exeter Medical School, University of Exeter, Exeter, UK. [38]Department of Pathology, Beth Israel Deaconess Medical Center, Harvard Medical School, Boston, USA. [39]Departments of Radiology, Medicine, and Psychiatry, University of California-San Francisco, San Francisco, USA. [40]Department of Veterans Affairs Medical Center, San Francisco, USA. [41]These authors contributed equally: Kwangsik Nho, Shannon L. Risacher. *Lists of authors and their affiliations appear at the end of the paper.
✉e-mail: asaykin@iu.edu

## for the Alzheimer's Disease Neuroimaging Initiative (ADNI)

Kwangsik Nho[1,2,3,4,41], Liana G. Apostolova[1,3,5,6], Kelley Faber [6,7], Martin R. Farlow [3,5], Tatiana Foroud [3,6,7], Paul Aisen[9], Reisa Sperling [10], Clifford R. Jack Jr. [12], William J. Jagust [13], Susan Landau[13], Ronald C. Petersen [25], Prashanthi Vemuri [12], Oscar L. Lopez [19,29], Michael W. Weiner[39,40] & Andrew J. Saykin [1,2,3,5,6] ✉

## the Department of Defense Alzheimer's Disease Neuroimaging Initiative (DoD-ADNI)

Paul Aisen[9], Ronald C. Petersen [25] & Michael W. Weiner[39,40]

## the Anti-Amyloid Treatment in Asymptomatic Alzheimer's Study (A4 Study) and Longitudinal Evaluation of Amyloid Risk and Neurodegeneration (LEARN)

Jared R. Brosch [3,5], Martin R. Farlow [3,5], Paul Aisen[9], Reisa Sperling [10], Sergey Shcherbinin[11], Clifford R. Jack Jr. [12], Colin L. Masters[18], Elizabeth Mormino[22], Rachel F. Buckley [10], Keith Johnson[10,23], Oscar L. Lopez [19,29] & Tammie Benzinger[32]

## the Australian Imaging, Biomarker & Lifestyle Study (AIBL)

Simon M. Laws [17], Tenielle Porter [17], Colin L. Masters[18] & Christopher C. Rowe [16,18]

