## [Peer Review File · Nature Communications]

CYP1B1-RMDN2 Alzheimer's disease endophenotype locus identified for cerebral tau PETEditorial Note: This manuscript has been previously reviewed at another journal that is not operating a transparent peer review scheme. This document only contains reviewer comments and rebuttal letters for versions considered at *Nature Communications*.

REVIEWER COMMENTS

Reviewer #1 (Remarks to the Author):

This is a review of the revised manuscript submitted by Nho, Risacher et al., in which they performed a GWAS on cortical tau PET signal using data from multiple cohorts. They identified a novel CYP1B1-RMDN2 Alzheimer's disease locus and elaborated on this finding by performing several additional analyses/experiments. Together with the relatively large sample size of the study, this thorough investigation of the top GWAS hit(s) in tissue and mouse models is the main strength of this work. Overall, this is important and timely work. The authors have addressed most of the previous comments adequately and provided additional data to support their arguments.

There are a few comments left, that I would recommend some further action.

Previous comment #1: Please add the description of the ROIs (or at least the cortical one) to the main text, as knowing the characteristics of the phenotype for the genetic analyses helps the reader understanding the results.

Previous comment #6: I agree with the author response below, please mention this in the main text as well so that the reader does not overinterpret that specific finding.

"We are not claiming tau specificity of spatial gene expression. We present the Allen Human Brain Atlas data to support that these genes are expressed in relevant brain regions but not with any exclusivity. The AHBA did not include patients with Alzheimer's disease, which limits their utility for testing disease related hypotheses."

Previous comment #8: I again fully agree with the author response here and would encourage them to add this as a limitation/future direction to the main manuscript.

Previous comment #11: Follow-up data (similar to ADNI) are available for most of the cohorts included in this manuscript. It should be acknowledged that conducting follow-up analyses in sub-cohorts of the entire study population is a limitation of the study.

Thank you for the thorough responses to most of the other comments.

Reviewer #2 (Remarks to the Author):

This revised version responds to the comments of three independent reviewers. Many of the comments have been addressed. However, I would like to highlight some points that need to be better described/discussed:

Reviewer 1 and Reviewer 2 - comments on CSF tau/p-tau concentrations. As requested by reviewer 1, the authors were happy to provide pTau181/SNP associations for ADNI and ADNI-DoD samples. These results are advantageously significant in these small datasets. Reviewer 2 asked for an extension of the tau results using the currently largest GWAS on CSF biomarkers. Most of the results were inconclusive and the authors decided not to show these data for two main reasons:

- Only summary statistics on p-Tau are available in Jansen et al. However, tau and p-Tau are highly correlated and it is not clear why similar data generated from ADNI and ADNI-DoD samples (n=525) are included in the manuscript rather than the ones generated from the EADB GWAS (n=8000).
- The authors argue that "CSF biomarkers, although associated with PET, are likely to measure different properties, forms or states of abnormal proteins in AD/ADRD". This is clearly possible, but so is the analysis they originally presented. If they do not want to show the data generated by Jansen et

al. based on this argument, all CSF data should be removed from the manuscript to be consistent.

Reviewer 1 asked for a forest plot and she/he is right. The authors should also independently present the results of the discovery and replication meta-analyses in their figure.

Reviewer 1 also asked why the number of PCs used for adjustments differed between studies. The answer is not really satisfactory as it is expected to keep those that remain significant in the statistical model. Keeping only PC1 and PC2 may be correct, but the wording is inappropriate: what does the phrase "were judged" mean?

Reviewer 2 and reviewer 3 asked for data on the association of genes/SNPs with the risk of AD. The results provided in this response are clearly negative and this should be clearly stated in the paper. The argument used by the authors is incorrect and therefore the title should be changed accordingly. There is no reason to state that this locus is an Alzheimer's disease locus. According to the results presented, it is a locus associated with the tau-pet endophenotype.

Reviewer #4 (Remarks to the Author):

Reviewer 3 mentioned that the authors should discuss the presence or absence of associations of these SNPs to AD risk. The authors showed the table in the responses. I found this sentence "the two SNPs were not significantly associated with AD in recent large-scale AD GWAS summary statistics" in lines 266 -267. I agree with the Reviewer 3's comment. The authors should discuss more details including the effect sizes and/or p-values so that readers could recognize that the findings are novel, or should cite the publications at least.

I agree with the Reviewer 3's comment about the gene-sets analyses. In the response, the authors mentioned: "including those that regulate reactive oxygen species, metabolic processes, monooxygenase activity, transferase activity, and the transfer of hexosyl groups". But in the Discussion, the authors described "MHC, postsynaptic membrane, postsynaptic density, synapse organization, and calcium channel activity". I also did not see the connection between the gene-sets analyses and the CYP1B1-RMDN2 locus. The authors should show which pathways included the CYP1B1-RMDN2 locus, and what genes were involved in the top pathways.

The authors should add what they mentioned about the APOE adjustment in the text (not just in the response).

Manuscript NCOMMS-23-54688A

Nho, Risacher et al Response to Reviewers' Comments:

We sincerely thank the three current and one prior reviewer and Editor for their thoughtful comments and detailed recommendations, all of which have helped to improve the clarity of our report. Below we address each of the Reviewer's comments.

Reviewer #1:

This is a review of the revised manuscript submitted by Nho, Risacher et al., in which they performed a GWAS on cortical tau PET signal using data from multiple cohorts. They identified a novel CYP1B1-RMDN2 Alzheimer's disease locus and elaborated on this finding by performing several additional analyses/experiments. Together with the relatively large sample size of the study, this thorough investigation of the top GWAS hit(s) in tissue and mouse models is the main strength of this work. Overall, this is important and timely work. The authors have addressed most of the previous comments adequately and provided additional data to support their arguments.

There are a few comments left, that I would recommend some further action.

- Previous comment #1: Please add the description of the ROIs (or at least the cortical one) to the main text, as knowing the characteristics of the phenotype for the genetic analyses helps the reader understanding the results.

Response: The cortical ROI included all regions from the Freesurfer parcellation (aparc), which is based on the Desikan-Killiany atlas. We added text and a citation to the supplemental section (Page 2) as follows:

“..., which included all regions from the Freesurfer parcellation (aparc) based on the Desikan-Killiany atlas...”

- Previous comment #6: I agree with the author response below, please mention this in the main text as well so that the reader does not overinterpret that specific finding.

Response: We agree with the reviewer and added a comment about the AHBA findings in the limitations/future direction section of the discussion (Page 17) as follows:

“The Allen Human Brain Atlas results suggest that the genes identified in this analysis are expressed in tau-relevant brain regions. However, these findings do not indicate that expression of these genes is exclusive to brain regions with high tau. Notably, the

AHBA did not include patients with ADRD, which limits their utility for disease-related hypotheses.”

- Previous comment #8: I again fully agree with the author response here and would encourage them to add this as a limitation/future direction to the main manuscript.

Response: We agree with the reviewer and added the following text to the limitations/future direction section of the discussion (Pages 16-17):

“...Although sex differences are increasingly recognized as important for precision medicine in ADRD, the current study was not designed or powered to thoroughly test these effects. Future studies that assess the presence and pattern of sex differences in longitudinal studies with larger samples are warranted...”

- Previous comment #11: Follow-up data (similarly to ADNI) are available for most of the cohorts included in this manuscript. It should be acknowledged that conducting follow-up analyses in sub-cohorts of the entire study population is a limitation of the study.

Response: We agree with the reviewer and added the following text to the limitations/future direction section of the discussion (Page 17):

“...Even though a number of the cohorts included in the present manuscript have longitudinal follow-up, the current study focused primarily on cross-sectional associations. Future studies to evaluate longitudinal follow-up in these cohorts, including analyses of longitudinal tau PET phenotypes, are also warranted...”

Reviewer #2:

This revised version responds to the comments of three independent reviewers. Many of the comments have been addressed. However, I would like to highlight some points that need to be better described/discussed:

- Reviewer 1 and Reviewer 2 - comments on CSF tau/p-tau concentrations. As requested by Reviewer 1, the authors were happy to provide pTau181/SNP associations for ADNI and ADNI-DoD samples. These results are advantageously significant in these small datasets. Reviewer 2 asked for an extension of the tau results using the currently largest GWAS on CSF biomarkers. Most of the results were inconclusive and the authors decided not to show these data for two main reasons: Only summary statistics on p-Tau are available in Jansen et al. However, tau and p-Tau are highly correlated, and it is not clear why similar data generated from ADNI and ADNI-DoD samples (n=525) are included in the manuscript rather than the ones generated from the EADB GWAS (n=8000). The authors argue that "CSF

biomarkers, although associated with PET, are likely to measure different properties, forms or states of abnormal proteins in AD/ADRD". This is clearly possible, but so is the analysis they originally presented. If they do not want to show the data generated by Jansen et al. based on this argument, all CSF data should be removed from the manuscript to be consistent.

Response: Following the reviewer’s suggestion, we included the results from large-scale GWAS for CSF biomarkers in the Results section as follows:

“We reviewed the GWAS summary statistics from two large-scale GWAS for CSF biomarkers.^{14,21} rs1478361 was associated with CSF total-tau levels but not CSF p-Tau levels. rs1478361, which is in strong LD with rs2113389 ($r^2 = 0.96$ and $D' = 1.00$), was associated with CSF total tau levels ($n=3,076$; $\beta=0.0176$; $p\text{-value}=0.0295$).¹⁴ Within the CYP1B1 locus, the most significant SNPs for CSF p-Tau levels were rs12463523 ($p\text{-value}=0.0026$) from the Deming et al. paper¹⁴ and rs9341266 ($p\text{-value}=0.0029$) from the Jansen et al. paper.²¹”

- Reviewer 1 asked for a forest plot and she/he is right. The authors should also independently present the results of the discovery and replication meta-analyses in their figure.

Response: Following the reviewer’s suggestion, we provided forest plots for the discovery and replication datasets, separately, as follows:

a) Discovery dataset

b) Replication dataset

c) Combined dataset

- Reviewer 1 also asked why the number of PCs used for adjustments differed between studies. The answer is not really satisfactory as it is expected to keep those that remain significant in the statistical model. Keeping only PC1 and PC2 may be correct, but the wording is inappropriate: what does the phrase "were judged" mean?

Response: In the analysis, we selected only non-Hispanic participants of European ancestry that clustered with HapMap CEU (Utah residents with Northern and Western European ancestry from the Centre d'Etude du Polymorphisme Humain collection) or TSI (Toscans in Italy) populations based on multidimensional scaling analysis using GWAS genotyping data. This may explain why none of the first four PCs significantly influenced cortical tau deposition. To be conservative in controlling potential subpopulation stratification effects, while avoiding introduction of statistical error due to inclusion of nonsignificant covariates, we decided to retain PC1 and PC2 as covariates.

- Reviewer 2 and Reviewer 3 asked for data on the association of genes/SNPs with the risk of AD. The results provided in this response are clearly negative and this should be clearly stated in the paper. The argument used by the authors is incorrect and therefore the title should be changed accordingly. There is no reason to state that this locus is an Alzheimer's disease locus. According to the results presented, it is a locus associated with the tau-pet endophenotype.

Response: In the Discussion section, we clearly described the results from large-scale AD GWAS and cited the publications as follows:

“However, recent large-scale AD GWAS studies have shown that the two SNPs were not significantly associated with AD with different association directions across the studies (p -value > 0.05).^{4,22-24} This lack of significant association may reflect heterogeneity in case-control ascertainment based on clinical diagnosis and is consistent with a selective association elucidated using quantitative endophenotype analysis”.

In addition, we appreciate the reviewer’s nuanced point regarding the title and have changed it to be more precise. The new title is “Novel CYP1B1-RMDN2 Alzheimer’s disease endophenotype locus identified for cerebral tau PET.”

Reviewer #4:

- Reviewer 3 mentioned that the authors should discuss the presence or absence of associations of these SNPs to AD risk. The authors showed the table in the responses. I found this sentence “the two SNPs were not significantly associated with AD in recent large-scale AD GWAS summary statistics” in lines 266 -267. I agree with the Reviewer 3’s comment. The authors should discuss more details including the effect sizes and/or p-values so that readers could recognize that the findings are novel or should cite the publications at least.

Response: In the Discussion section, we clearly described the results from large-scale AD GWAS and cited the publications as follows:

“However, recent large-scale AD GWAS studies have shown that the two SNPs were not significantly associated with AD with different association directions across the studies (p -value > 0.05).^{4,22-24} This lack of significant association may reflect heterogeneity in case-control ascertainment based on clinical diagnosis and is consistent with a selective association elucidated using quantitative endophenotype analysis”.

- I agree with the Reviewer 3’s comment about the gene-sets analyses. In the response, the authors mentioned: “including those that regulate reactive oxygen species, metabolic processes, monooxygenase activity, transferase activity, and the transfer of hexosyl groups”. But in the Discussion, the authors described “MHC, postsynaptic membrane, postsynaptic density, synapse organization, and calcium channel activity”. I also did not see the connection between the gene-sets analyses and the CYP1B1-RMDN2 locus. The authors should show which pathways included the CYP1B1-RMDN2 locus, and what genes were involved in the top pathways.

Response: We performed gene-set enrichment analysis using GWAS summary statistics to identify pathways and functional gene sets associated with cortical tau deposition to gain a deeper biological understanding. These results are complementary to our primary findings and may help to inform future studies. GO terms for several pathways containing genes near the *CYP1B1* locus were significant, including those that regulate reactive oxygen species, metabolic processes, monooxygenase activity, Golgi organization, and endoplasmic reticulum organization. In addition, the KEGG pathway for steroid hormone biosynthesis was significant.

Following the reviewer's suggestions, we added the following in the Results section and provided gene lists involved in the top pathways in the Supplementary materials.

“GO terms for several pathways containing genes near the CYP1B1 locus were significant, including those that regulate reactive oxygen species, metabolic processes, monooxygenase activity, Golgi organization, and endoplasmic reticulum organization, as well as the KEGG pathway for steroid hormone biosynthesis.”

- The authors should add what they mentioned about the APOE adjustment in the text (not just in the response).

Response: A sentence was added to Statistical Analysis in the Methods section (page 19).

“APOE4 status was included as a covariate because its effect was modeled to understand the contribution of the discovered CYP1B1-RMDN2 locus above and beyond APOE4 and to assess whether there is epistasis with APOE4.”

REVIEWER COMMENTS

Reviewer #4 (Remarks to the Author):

The authors addressed the Reveiwer 3 concerns accordingly.

Manuscript NCOMMS-23-54688B

Nho, Risacher et al Response to Reviewers' Comments:

We sincerely thank the reviewers and the Editor for their thoughtful past and current comments and detailed recommendations, all of which have helped to improve the clarity of our report. Below we have added text suggested by the Editor in response to Reviewer #1 and Reviewer #2's comments.

Reviewer #1:

- Please include the description of the cortical ROI in the main text (as per Reviewer #1's previous request), as well as in the supplementary information.

Response: In addition to being provided in the Supplemental text, a description of the cortical ROI has been added to the Methods section of the main text (page 19).

“Genome-wide association analysis (GWAS): Cortical tau deposition (*the weighted average SUVR of all cortical regions from FreeSurfer version 6.1 parcellation (aparc)*) followed a normal distribution after a rank-based inverse normal transformation.”

Reviewer #2:

- Reviewer #2 indicated privately that they appreciated the addition of the forest plot, but noted that a heterogeneity test is missing, and that if the I² is significant, a random meta-analysis should be presented. They feel that this is particularly true at the replication stage. We ask that you address this point.

Response: A heterogeneity analysis in METAL was performed to evaluate the possible effect of study heterogeneity on the results. The heterogeneity test for rs2113389 yielded Heterogeneity $I^2=27.8$ and Heterogeneity p -value= 0.217 for the discovery sample and Heterogeneity $I^2=52.0$ and Heterogeneity p -value =0.080 for the replication sample.

The results have been added to the Results section (page 9) of the main text, “The minor allele T of rs2113389 (MAF=0.146) was associated with higher tau (Z score=5.68; p -value= 1.37×10^{-8} ; **Heterogeneity $I^2=27.8$; Heterogeneity p -value= 2.17×10^{-1}**). A replication meta-analysis in five additional cohorts (n=1,600) showed that the significant SNPs (rs2113389 and rs918804) in the discovery stage were replicated with the same association direction (Z score=3.83, p -value= 1.26×10^{-4} , **Heterogeneity $I^2=52.0$,**

Heterogeneity p-value=8.02 x 10⁻²; Z-score=-2.97, p-value=2.97x10⁻³, Heterogeneity I²=59.5; Heterogeneity p-value=5.99 x 10⁻², respectively; Supplemental Figure 1).

Also, a description was added to the Methods section (page 19) of the main text, “***A fixed effect meta-analysis with an inverse variance weighted approach was performed using METAL, and a heterogeneity analysis in METAL was performed to evaluate the possible effect of study heterogeneity on the results.***”

The results were also added to the Supplementary Figure 1 (Forest plots; pages 28-29) of the Supplemental text, “In the discovery sample (ADNI, A05, IMAS, ADNI-DoD, HABS, UPitt ADRC), the minor allele T of rs2113389 (MAF=0.146) was associated with higher tau (Z score=5.68; p-value=1.37 x 10⁻⁸; ***Heterogeneity I²=27.8; Heterogeneity p-value=2.17 x 10⁻¹***). We conducted a replication meta-analysis in five additional cohorts (n=1,687). The top SNP (rs2113389) in the discovery stage was replicated with the same association direction (Z score=3.83; p-value=1.26 x 10⁻⁴; ***Heterogeneity I²=52.0, Heterogeneity p-value=8.02 x 10⁻²***). The overall meta-analysis showed a genome-wide significant association of rs2113389 with cortical tau deposition.”

REVIEWERS' COMMENTS

Reviewer #2 (Remarks to the Author):

The authors answered to my comment